# Impact of Testicular Cancer on the Socio-Economic Health, Sexual Health, and Fertility of Survivors—A Questionnaire Based Survey

**DOI:** 10.3390/cancers17111826

**Published:** 2025-05-30

**Authors:** M. Raheel Khan, Patrice Kearney Sheehan, Ashley Bazin, Christine Leonard, Lynda Corrigan, Ray McDermott

**Affiliations:** 1School of Medicine, University College Dublin, D04 V1W8 Dublin, Ireland; ray.mcdermott@tuh.ie; 2Tallaght University Hospital, D24 NR0A Dublin, Ireland; patrice.kearneysheehan@tuh.ie (P.K.S.); ashley.bazin@tuh.ie (A.B.); christine.leonard@tuh.ie (C.L.); lynda.corrigan@tuh.ie (L.C.); 3St Vincent’s University Hospital, D04 T6F4 Dublin, Ireland

**Keywords:** testicular cancer, survivors, socio-economic impact, germ cell tumours, cancer survivors, sexual health

## Abstract

Testicular cancer is the most common cancer in young males, with high cure rates. The treatment of testicular cancer involves surgery, chemotherapy, and radiotherapy, which can cause late side-effects in survivors. We performed this study in Ireland to assess the impact of cancer diagnosis and its treatment on testicular cancer survivors. We requested that the participants answer an anonymised questionnaire designed to estimate the effect of testicular cancer diagnosis and treatment on their social, economic, and sexual health. Fertility complications were also included in the survey. Almost half of the 83 participants reported some effect on performance at work and personal finances, while one third of them reported issues with job security and relationships with partners. Similar results were seen with sexual health, but no major deterioration in fertility was observed in survivors. Survivorship care should provide holistic care to patients, considering their economic, social, and sexual health.

## 1. Introduction

Testicular cancer (TC) is a rare tumour (1%) that is mostly diagnosed at a younger age (15 to 35 years) [1]. Since the advent of the cisplatin-based chemotherapy regimens, the cure rates are reported to be above 95% [2,3]. Cisplatin-based multi-drug regimens, first used in 1974, hugely improved the prognosis compared to the pre-cisplatin era [4,5]. Moreover, recent decades have seen an increase in the incidence along with an improvement in the survival rate [1]. However, these high cure rates come at a cost of late side-effects, particularly in the patients who receive chemotherapy and radiotherapy [6,7]. These side-effects encompass a list of malignant and non-malignant conditions including late recurrences, second malignant neoplasms, cardiovascular disease, renal insufficiency, pulmonary disease, hypogonadism, infertility, and metabolic syndrome [8,9]. On top of these complications, a higher incidence of psychological conditions including anxiety and fear of cancer recurrence has also been reported in this cohort [10,11]. As most of the survivors are young at the time of diagnosis, they have to brave these complications for decades to come. These physical and psychological complications have been reported to cause a negative impact on the social, economic, and sexual well-being of TC survivors [12,13]. Higher levels of disability, unemployment, and sick leave have been reported in recent studies [12].

An estimated half of patients have abnormal semen analysis at the time of diagnosis before any treatments [14]. Chemotherapy, radiotherapy, and surgeries can further deteriorate sperm production, motility, and erectile function [15,16]. As a result, a reduced rate of paternity is found in TC survivors compared to the normal population, even with pre-treatment semen cryopreservation techniques [15]. These consequences of treatment also cause the re-entry and readjustment problems for the survivors back into society, as they are nervous and anxious about their fertility [17]. Many researchers have reported long-term deterioration in sexual health, including erectile dysfunction, sexual dissatisfaction, low sex drive, and orgasmic dysfunction [18]. This long-term deterioration in sexual health results from multiple factors, including hypogonadism, anxiety, systemic diseases, impaired body image perception, and a sense of loss of masculinity [18]. These complications, in turn, have a significant effect on the social lives of TC survivors, particularly relationships with their partners. Spouses and partners, especially those who began their relationship after TC diagnosis, are poorly affected in terms of their quality of life and stress response symptoms [19]. In addition, long-term side-effects on career and financial well-being are also reported in TC survivors. Jobs and careers can be affected by late comorbidities like peripheral neuropathy and chronic cancer-related fatigue [20]. A systematic review by Schepisi et al. conveyed a concerning outlook of survivors, revealing a higher incidence of joblessness, absenteeism, and neuroticism, especially in those involved in manual labour [17]. TC diagnosis and treatments, mainly chemotherapy, have been found to cause financial distress and financial burden on survivors [13].

In the last few years, we have seen growing interest in cancer survivorship, with numerous studies performed to look at different aspects of survivorship care. Previously, we investigated the prevalence of mortality and morbidity in TC survivors attending Tallaght University Hospital in Dublin, Ireland [17,18]. Now, this study was conducted to provide an estimation of the socio-economic, sexual, and fertility-related difficulties faced by TC survivors in Ireland. As we believe a multi-team, collaborative model of care will be key to improving the quality of life in survivors, our study aims to identify areas in need of further research and quality improvement initiatives. Our study may also be helpful for other urban populations in the UK and Western Europe.

This article is a revised and expanded version of a paper entitled “Impact of testicular cancer on socio-economic and sexual health of survivors: A questionnaire-based survey”, which was presented at ESMO Sarcoma and Rare Cancer Congress, held in Lugano, Switzerland, on 14 March 2024 [21].

## 2. Materials and Methods

### 2.1. Study Design and Participants

We requested the patients attending the survivorship clinic in a tertiary care cancer centre in Ireland to participate in the study. A self-reported, completely anonymised survey questionnaire (provided in Appendix A) was filled out by the participants after verbal consent. The questionnaire was developed on an ad hoc basis and not previously validated. All patients who were requested agreed to participate. This study was completely anonymised to encourage participation. Based on our clinic parameters, all participants were known cases of testicular cancer who had successfully completed at least five years of surveillance after treatment.

### 2.2. Data Assessment

We designed the questionnaire to assess the self-reported impact of testicular cancer and treatment on the fertility, sexual health, and socio-economic aspects, including job and careers, of TC survivors.

To assess the socio-economic impact, patients were asked to report the negative impact of cancer on their performance/productivity at work, job stability, career choices, personal financial goals, and the relationship with their partner. The responses were recorded as one of five options on a five-point Likert scale, including no effect, minor effect, moderate effect, significant effect, and very significant effect.

Similar responses were recorded to assess sexual health, where the patients were asked to report the extent to which the cancer and the treatment affected their body image perception, ability to find a new partner, libido, erectile function, ejaculation, and satisfaction with sexual activity.

Infertility was defined as the inability to conceive after at least one year of unprotected intercourse with the partner. Participants were also asked if they attempted any medical procedures to assist fertility. We asked patients if they had any children before cancer and after cancer. Also, we inquired if they had any issues with fertility and if the issues lasted more than a year.

### 2.3. Statistical Analysis

All responses were recorded on the Excel-based (Microsoft Excel v16) spreadsheets. To analyse and report the responses, we calculated the percentages of total participants. Unanswered questions were reported as a blank response.

### 2.4. Ethics and Good Clinical Practice

This study was granted approval in full by JREC (Joint Research and Ethics Committee) of Tallaght University Hospital, Dublin. Survey questionnaires were completely anonymised to protect patient identities. All staff involved in the study were trained in ethics, data protection, and Good Clinical Practice. This study was conducted in compliance with EU and Irish ethics and data protection guidelines. Complete anonymity was maintained during data collection, processing, and analysis. Informed verbal consent was taken from participants since written consent was not required as the questionnaires were completely anonymised without any identifiers. All authors had complete access to the data and contributed to the conceptualisation, research methodology, writing, and editing of this study. The initial draft was written by the first and last author.

## 3. Results

This study was conducted from January 2023 to November 2023 in the testicular cancer survivorship clinic of Tallaght University Hospital, Dublin, Ireland. A total of 83 patients completed our survey questionnaire. All patients agreed to participate in this study. A total of 66 data points were left blank on the questionnaires out of 1262 total, with a response rate of 5%.

### 3.1. Socio-Economic Impact

To investigate the socio-economic impact on the testicular cancer survivors, we asked the participants about changes in five aspects of their livelihood since the diagnosis of cancer. These five parameters included performance/productivity at work, job security/stability, career choices, personal financial goals, and relationship with their partners (Figure 1). When asked about performance or productivity at work, 58% (47) reported no effect, 18% (15) reported a minor effect, 6% (5) reported a moderate effect, and 8% (7) reported a significant and very significant effect. Regarding their job security or stability, the responses showed 72% (59) reporting no effect, 6% reporting a minor effect, 7% reporting a moderate effect, 2% reporting a significant effect, and 10% reporting a very significant effect.

Moreover, we gauged the impact of cancer on their career choices as the majority of patients are diagnosed in the early years of their career. On one hand, 13% (11) reported the effect as moderate and 10% reported as very significant; on the other hand, 71% (58) of patients reported no effect on their careers. With respect to the financial implications of cancer diagnosis and treatment, 57% of responders noted no effect on their personal financial goals. On the contrary, 11% reported very significant, 4% significant, 14% moderate, and 13% minor effects.

Participants were asked if cancer had any effect on their relationship with their partners. Almost two-thirds (64%) reported no effect, while the rest reported some effect. Of those who reported some effect, 27% reported a minor effect, 33% reported a moderate effect, 19% reported a significant effect, and 16% reported a very significant effect. Nearly all (96%) of the participants responded to the question asking about any difficulty in finding a new partner since the diagnosis of cancer. The responses revealed that 80% (66) of respondents did not notice any issues in this regard. Among those who experienced any difference, 43% had a minor effect, 31% had a moderate effect, 31% had a significant effect, while only 12% had a very significant effect.

### 3.2. Sexual Health

In this survey, the impact on sexual health was assessed in five different categories, including libido, erectile function, ejaculation, satisfaction with sexual activity, and body image perception (Figure 2). Almost half of our patients (54%) reported no effect on their libido. Although the rest of the participants stated otherwise, with 6% stating a very significant effect, 14% a significant effect, 12% a moderate effect, and 13% a minor effect. Erectile function reportedly remained intact in more than two-thirds of the respondents (70%), while it was affected very significantly in 6% and significantly in 10%. Moreover, 6% reported a minor and 8% a moderate effect. Similar to erectile function, ejaculation remained intact in 71% of patients. Conversely, 11% described the impact as very significant, 4% described the impact as significant, 8% described the impact as moderate, and 5% described the impact as minor. Over one-third (39%) of our testicular cancer survivors revealed an adverse impact on satisfaction with sexual activity. Of these, the majority stated only a minor-to-moderate effect (54%), and the rest (46%) experienced a significant or very significant effect. Almost half of patients (53%) reported that cancer no effect on their body image perception. At the same time, it was affected to a minor extent in 18%, a moderate extent in 17%, a significant extent in 10%, and a very significant extent in 2% of participants.

### 3.3. Fertility

One third of the participants experienced biological parenthood after the diagnosis of cancer, whereas two thirds of them reported no children since the testicular cancer diagnosis and treatment (Figure 3). It is important to note that nearly half of them (41%) had children before the diagnosis as per survey responses. Only a minority (17%) of the participants reported having trouble with conception; however, 63% reported no issues and 20% chose not to respond. When asked if the fertility issues stayed for more than a year, 54% of them responded in the negative, but 14% were affirmative. Approximately one-third of the respondents (31%) refrained from providing an answer to the question. We asked our participants if they sought any medical assistance for fertility issues; only 17% of them stated yes, while 67% stated no and 16% chose not to answer.

## 4. Discussion

The majority of our patients reported no major effect on their sexual and socio-economic health. Even the fertility rates after TC diagnosis and treatment were similar to the general population, as discussed below.

In this study, 6% of patients reported a very significant effect on their relationship with their partner. These findings are comparable to 5–10% divorce and break-up rates reported in a literature review by Schepisi et al. [17]. Another similar study at Yale University, USA, reported on the subjective impact of cancer on personal finances in survivors [22]. A total of 8.6% of participants reported impact as “a lot,” compared to the 11% in our study who reported the effect as “very significant”. However, the study at Yale was not confined to TC and included all cancer survivors.

Regarding the effect on career and job, 8–10% of responders revealed a very significant effect. Again, these findings are comparable to similar studies performed previously [12,22]. For instance, a major survey-based study conducted in Texas, USA, encompassing 4363 cancer survivors, revealed that 8.5% of survivors considered themselves unable to work since diagnosis [23]. This study included all cancer survivors, yet the results are similar to those in this study. Another study reported 67% of cancer survivors working full-time jobs after 5–7 years post-diagnosis, although some of them reported difficulties in carrying out their responsibilities [24]. Kerns et al. also reported similar outcomes after studying 1815 TC survivors from the US, the UK, and Canada in 2020 [12]. As many as 10% of the TC survivors were reported to be out of work, with a higher risk of unemployment compared to the age-adjusted general population (odds ratio: 2.67) [12]. They also reported a relatively higher impact on survivors who received chemotherapy [12].

As most TC survivors are young at the age of diagnosis [1], body image perception has a major influence on their emotional, sexual, and social well-being [25]. Rossen et al. reported that 17% of TC survivors in Denmark experienced altered body image, which was associated with all parameters of sexual dysfunction as well [25]. However, among our cohort of survivors, only 53% reported no effect, while the rest reported some degree of effect. In the same study, 24.4% of survivors reported reduced sexual interest, which is comparable to our finding of 20% reporting a significant and very significant effect on libido [25]. A similar trend was seen with respect to erectile function and satisfaction with sexual activity. Our results show 18% reporting a significant or very significant effect on erectile function compared to the 17% reported by Rossen et. al. With regards to satisfaction with sexual activity, 18% of our participants disclosed a major effect, compared to 14% in their study. Our participants’ responses on the Likert scale regarding ejaculation shows a worse impact than reported in other studies. Among the respondents, 15% described the effect as significant or very significant, whereas only 7% reported ejaculatory problems in the survey discussed above [25]. However, a systemic review by Schesipi et. al. estimates the proportion as between 29 and 44% [17]. In TC survivors, ejaculatory problems are mostly related to retroperitoneal lymph node dissection, and outcomes can be improved with nerve sparing techniques [26]. This variation could possibly be due to different trends in the surgical treatments in different countries and cancer centres.

The majority (80%) of patients reported no difficulties in finding a new partner, but unfortunately, our survey failed to identify participants already in stable relationships.

Spermon MD reported paternity after cancer in Norwegian TC survivors as 43% in 2003 [27], which is higher than the findings of this study (34%). This difference could be due to the general decline in fertility rates witnessed in Europe in the last couple of decades [28]. This study is not able to identify participants who were not planning to have any children. Also, a better overview can be obtained through more studies comparing paternity rates within different treatment modalities. The percentage of patients seeking medical assistance for fertility (17%) in this study is the same as reported by Spermon. The infertility rate (failure to conceive for more than 1 year) reported in our cohort of TC survivors (17%) is closer to the general population (16%) [28].

Participation was notably comprehensive, as every survivor contacted agreed to be involved. We believe one of the main reasons for the high recruitment was the complete anonymisation of questionnaires. However, this anonymisation also proved to be a major limitation of our study. Due to anonymisation, we were not able to stratify our data based on age at diagnosis, treatment types, stage of cancer at diagnosis, co-morbidities, and social parameters, e.g., employment status, relationship status, and education level. This stratification could have been very useful in identifying the risk factors for each of the major implications in the survivorship phase. The size of the cohort in this study was small; a multi-centre study with a larger cohort of patients could give us more significant and impactful data. With the small sample size and single-institute design, the results of this study have limited generalisability.

The participants were asked to reflect on their experiences over a period of several years, which makes them vulnerable to recall bias. This bias could have led to a significant over-estimation of the impact. Also, only the survivors attending the survivorship clinic were included in this study, which may lead to a selection bias.

Another considerable limitation comes from the absence of any comparative cohort as a control arm. A comparable control arm of the general population could help us to better define the effect of testicular cancer on these parameters.

The questionnaire we used in this study was developed on an ad hoc basis without previous validation. Our questionnaire lacks the authenticity and dependability of a validated questionnaire. We believe a comprehensive validated questionnaire to assess these aspects of survivorship care should be developed and used for future studies. Currently available questionnaires (e.g., EORTC QLQ-TC26) lack the focus on late survivorship care [29,30].

Given the small scale of this study and the limitations described above, we strongly believe that more research in this area is needed. We recommend multi-centre, pseudonymised, and case–control studies to acquire further insight into the subject of TC survivorship.

## 5. Conclusions

In this study, we have highlighted the issues being faced by testicular cancer survivors as a result of the late side-effects of cancer and its treatments. Although most of the TC survivors reported no significant effect on their socio-economic and sexual health, some of them struggled with the negative impacts. Regarding sexual health, most of the repercussions were felt in body image perception and libido, while social sequelae mainly included deterioration in one’s relationship with one’s partner. Economic health was affected in some TC survivors due to job instability and reduced performance at work. Survivorship care should attend to these aspects of well-being and should be designed to provide support where needed.

## Figures and Tables

**Figure 1 cancers-17-01826-f001:**
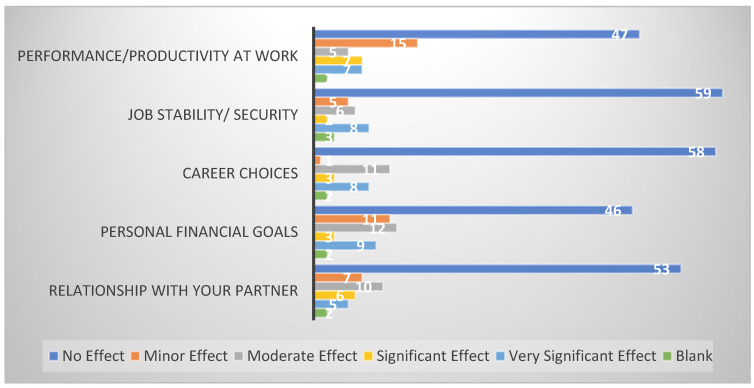
Socio-economic impact on testicular cancer survivors. Out of the 83 participants who responded on a 5-point Likert Scale, the chart displays the number of responses in each category.

**Figure 2 cancers-17-01826-f002:**
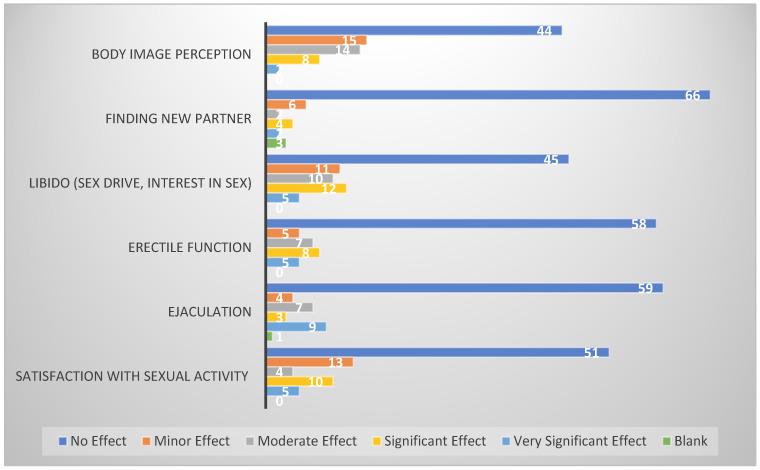
Impact on sexual health of testicular cancer survivors. Out of the 83 participants who responded on a 5-point Likert Scale, the chart displays the number of responses in each category.

**Figure 3 cancers-17-01826-f003:**
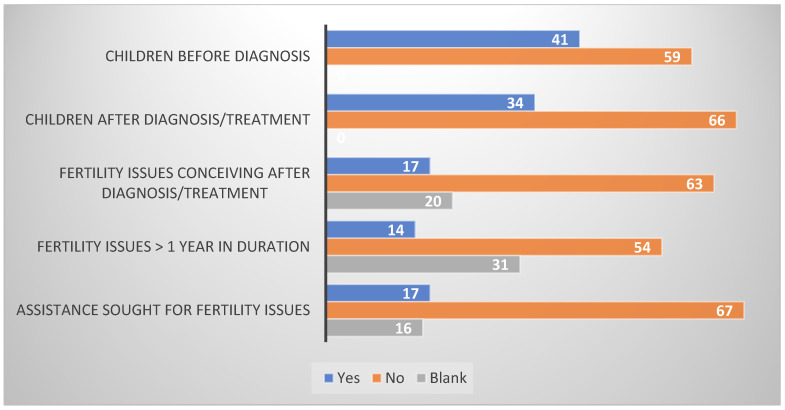
Fertility issues in testicular cancer survivors. Participants responded with “yes” or “no” to questions about fertility. The chart displays the number of responses and the number of blank responses.

## Data Availability

The original contributions presented in this study are included in the article/Appendix A. Further inquiries can be directed to the corresponding author.

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
