# Peer review of "Impact of Testicular Cancer on the Socio-Economic Health, Sexual Health, and Fertility of Survivors—A Questionnaire Based Survey"

_cancers, 2025, doi:10.3390/cancers17111826_

Round 1

Reviewer 1 Report

Comments and Suggestions for Authors

I congratulate the Authors for their study.

The only comment is that no information are given regarding tretments received by these patients (chemotherapy? number of cycles? salvage therapy?) I think that such infiormation are needed in this manuscript. Otherwise all patients seem alike .

Author Response

The only comment is that no information are given regarding treatments received by these patients (chemotherapy? number of cycles? salvage therapy?) I think that such information are needed in this manuscript. Otherwise all patients seem alike 

Authors’ Response: We appreciate the reviewers’ feedback, and we have mentioned this in the study limitations. Due to complete anonymisation, we couldn’t relate the questionnaire outcomes with different treatment modalities. We do understand that this is a major limitation of our study, and we hope further research is carried out to address this deficiency.

Reviewer 2 Report

Comments and Suggestions for Authors

Extremely interesting paper on evaluating the Impact of Testicular Cancer on Socio-Economic Health, Sexual Health and Fertility of Survivors - as doctors we sometimes concentrate on just treating a disease and forget about the other factors that might affect the quality of life from a psychological standpoint.

This paper evaluates important factors in cancer survivors using a a simple questionnaire that can be applied in any clinical practice.

Introduction chapter is very comprehensive, presents the main challenges that appear in patients that survive testicular cancer might endure - with an emphasis on the psychological strain that might be determined by the disease and the treatment.

Material and methods are clear, the study population and means of collection of data is presented in a very clear fashion. 

Results chapter presents the results of the questionnaire in a extensive manner.

Discussions are in line with the subject of the paper.

Conclusions chapter was clear: Although the majority of the patients presented no significant effect on the socio-economic and sexual health, there were some cases of repercussions of treatment on the patients' body image perception and libido which affected the patients' relationship with their partner and also economical health strain resulted from low productivity at work.

All these factors should be taken into account in testicular cancer patients and support should be offered where needed.  

I recommend for publication.

Author Response

Extremely interesting paper on evaluating the Impact of Testicular Cancer on Socio-Economic Health, Sexual Health and Fertility of Survivors - as doctors we sometimes concentrate on just treating a disease and forget about the other factors that might affect the quality of life from a psychological standpoint.

This paper evaluates important factors in cancer survivors using a simple questionnaire that can be applied in any clinical practice.

Introduction chapter is very comprehensive, presents the main challenges that appear in patients that survive testicular cancer might endure - with an emphasis on the psychological strain that might be determined by the disease and the treatment.

Material and methods are clear, the study population and means of collection of data is presented in a very clear fashion. 

Results chapter presents the results of the questionnaire in an extensive manner.

Discussions are in line with the subject of the paper.

Conclusions chapter was clear: Although the majority of the patients presented no significant effect on the socio-economic and sexual health, there were some cases of repercussions of treatment on the patients' body image perception and libido which affected the patients' relationship with their partner and also economical health strain resulted from low productivity at work.

All these factors should be taken into account in testicular cancer patients and support should be offered where needed.  

I recommend for publication.

Authors’ Response: We highly appreciate the positive feedback from the reviewer. We hope this paper can lead to further research and highlight the challenges faced by survivors.

Reviewer 3 Report

Comments and Suggestions for Authors

The topic is interesting; however, as your limitations state, there is a missed opportunity in being able to be more granular in the analyses due to the complete anonymisation. Perhaps asking the participants' age at diagnosis could have provided some insights, and I am unsure if this was intentionally omitted for anonymity. Other questions of interest such as whether a prosthesis was used and whether that had any impact on body image etc could have also been included. The paper is, overall, acceptably written. Please address the following:

  1. Line 51: I would not use the word "unfortunately". It is very fortunate that we are curing these young men of cancer. Perhaps "however".
  2. Line 79: "painted a grim picture" is overly negative. Think of how cancer survivors will feel reading your paper.
  3. Figures 1 - 3: these should be self-sufficient. Clarify in the caption that the numbers represent the number of patients and also provide the total number of patients.  
  4. Line 252: the word "phenomenal" to describe participation of 83 patients is hyperbolic. Please temper down.
  5. Discussion: the order of the paragraphs seems rather haphazard. Consider following the same order as the results section - socioeconomic first then sexual health and fertility.
  6. There are several minor but distracting grammatical issues such as starting sentences with numbers and using abbreviations such as "didn't" which should be avoided in scientific writing. Please consider revising.
  7. Supplementary file: it should be "effect" not "affect".
  8. Please upload the questionnaire as a supplementary file.

Author Response

The topic is interesting; however, as your limitations state, there is a missed opportunity in being able to be more granular in the analyses due to the complete anonymisation. Perhaps asking the participants' age at diagnosis could have provided some insights, and I am unsure if this was intentionally omitted for anonymity. Other questions of interest such as whether a prosthesis was used and whether that had any impact on body image etc could have also been included. The paper is, overall, acceptably written. Please address the following:

  1. Line 51: I would not use the word "unfortunately". It is very fortunate that we are curing these young men of cancer. Perhaps "however".

Authors’ Response: We appreciate the feedback, amendments have been made in the revised manuscript as advised.

  1. Line 79: "painted a grim picture" is overly negative. Think of how cancer survivors will feel reading your paper.

Authors’ Response: We appreciate the feedback and agree with the comment. The phrase has been replaced by “conveyed a concerning outlook” in the revised manuscript.

  1. Figures 1 - 3: these should be self-sufficient. Clarify in the caption that the numbers represent the number of patients and also provide the total number of patients.  

Authors’ Response: We agree with the comments from the reviewer; the captions of these figures have been amended in the revised manuscript.

  1. Line 252: the word "phenomenal" to describe participation of 83 patients is hyperbolic. Please temper down.
    Authors’ Response: We agree with the feedback, the revised manuscript has been modified as advised.

  2. Discussion: the order of the paragraphs seems rather haphazard. Consider following the same order as the results section - socioeconomic first then sexual health and fertility.
    Authors’ Response: We agree with the feedback, revised manuscript has been modified as advised.

  3. There are several minor but distracting grammatical issues such as starting sentences with numbers and using abbreviations such as "didn't" which should be avoided in scientific writing. Please consider revising.

Authors’ Response: We highly appreciate the feedback. On thorough revision of the manuscript, all such instances have been addressed.

  1. Supplementary file: it should be "effect" not "affect".
    Authors’ Response: We are thankful to the reviewers for pointing it out. We have corrected the mistake in the revised submission. 

  2. Please upload the questionnaire as a supplementary file

Authors’ Response: Questionnaire now included in Supplementary files

Reviewer 4 Report

Comments and Suggestions for Authors

We thank the authors for their work that addresses a relevant survivorship issue in testicular cancer and adds regional data from Ireland. 

Introduction:

Streamline and re-organize paragraphs for better logical flow....Consider separating historical background from the rationale for the current study.

Methods:

Elaborate on the development and validation of the questionnaire. Indicate if items were based on validated instruments or developed ad hoc. Describe how Likert scale responses were analyzed (e.g., were any thresholds applied to dichotomize effect?)....Mention response rate and if any patients declined participation (even if the number is zero). Discuss handling of missing data? how many items were missing??

Results:

Figures are difficult to interpret; revise axis labels and legends to improve clarity. Clarify if there were any subgroup comparisons (e.g., based on treatment modality, time since diagnosis).

Discussion:

Acknowledge the limited generalizability due to the sample size and single-institution design. Consider discussing potential recall bias, especially given the retrospective nature and long follow-up period. The comparison to older international studies is appreciated, but regional differences should be emphasized.

Limitations:

Rightly acknowledged limitations, but discussion could benefit from deeper exploration of how these affect the interpretation of your results

Comments on the Quality of English Language

Language and Formatting:

Consider professional English editing for sentence clarity and grammar...Remove redundant phrases throughout the manuscript...Improve transitions...

Author Response

We thank the authors for their work that addresses a relevant survivorship issue in testicular cancer and adds regional data from Ireland. 

Introduction:

Streamline and re-organize paragraphs for better logical flow....Consider separating historical background from the rationale for the current study.

Authors’ Response: We highly appreciate the feedback. Amendments have been made in revised manuscript as advised.

Methods:

Elaborate on the development and validation of the questionnaire. Indicate if items were based on validated instruments or developed ad hoc.

Authors’ Response: Many thanks for valuable feedback. Our questionnaire was ad hoc and lacks validation, we have mentioned the details now in revised manuscript.

Describe how Likert scale responses were analyzed (e.g., were any thresholds applied to dichotomize effect?)....

Authors’ Response: We appreciate reviewer’s feedback. The responses were subjective, self-reported assessment by participants on a Likert Scale. We have attached the questionnaire in supplements which include five options (No Effect, Minor Effect, Moderate Effect, Significant Effect and Very Significant Effect). The response were recorded and reported without any further processing.  

Mention response rate and if any patients declined participation (even if the number is zero). Discuss handling of missing data? how many items were missing??

Authors’ Response: We agree with the reviewer’s comments. Missing data was reported as blank, mentioned already in methods under statistical analysis. Response rate and number of missing items are reported now in revised manuscript.

Results:

Figures are difficult to interpret; revise axis labels and legends to improve clarity.

Authors’ Response: We agree with the reviewer’s feedback. We have added more details in the caption in revised manuscript.

Clarify if there were any subgroup comparisons (e.g., based on treatment modality, time since diagnosis).

Authors’ Response: We are thankful for the valuable opinion. Due to complete anonymity, subgroup for comparisons couldn’t be established. We have mentioned this in study limitations.

Discussion:

Acknowledge the limited generalizability due to the sample size and single-institution design.

Authors’ Response: We agree with the feedback, revised manuscript now amended as advised.

Consider discussing potential recall bias, especially given the retrospective nature and long follow-up period.

Authors’ Response: Very important point raised by the reviewer, we have added this in limitation section of revised manuscript.

The comparison to older international studies is appreciated, but regional differences should be emphasized.

Authors’ Response: We agree with the reviewers opinion. Revised manuscript is amended as advised.  

Limitations:

Rightly acknowledged limitations, but discussion could benefit from deeper exploration of how these affect the interpretation of your results

Authors’ Response: Based on the valuable of opinion of reviewer, we have revised the section in this manuscript.

Reviewer 5 Report

Comments and Suggestions for Authors

Dear authors, after reading your paper, I think it has some drawbacks:

Patients "attending the survivorship clinic" were requested to participate. This is a form of convenience sampling. Patients attending this specific clinic might differ systematically from those who do not (e.g., more proactive about health, different socio-economic status, different severity of long-term effects), potentially introducing selection bias.

"All patients who were requested agreed to participate." While this seems optimistic, it's unusually high and might warrant further detail. For example, how many patients were approached in total? Were there any eligibility criteria applied before the request was made that might have screened out specific individuals? If indeed all approached patients participated, it would reduce non-response bias, but the initial selection method is still a factor.

The text states, "We designed the questionnaire..." but makes no mention of whether this questionnaire was validated (e.g., through pilot testing, cognitive interviewing, or comparison with established instruments). An unvalidated questionnaire may have issues with clarity, interpretability, reliability, or validity of the questions, potentially leading to inaccurate or biased responses.

Your study doesn't explore relationships between variables (e.g., does the impact on sexual health correlate with the effects on relationships? Do those with children pre-cancer report different socio-economic impacts?)

You should also provide the number of patients because only the percentages are provided; it is not clear if the number is from the total or from a subgroup.

Author Response

Dear authors, after reading your paper, I think it has some drawbacks:

Patients "attending the survivorship clinic" were requested to participate. This is a form of convenience sampling. Patients attending this specific clinic might differ systematically from those who do not (e.g., more proactive about health, different socio-economic status, different severity of long-term effects), potentially introducing selection bias.

Authors’ Response: We agree with the opinion of reviewer and this selection bias is now mentioned in limitations section of revised manuscript.

"All patients who were requested agreed to participate." While this seems optimistic, it's unusually high and might warrant further detail. For example, how many patients were approached in total? Were there any eligibility criteria applied before the request was made that might have screened out specific individuals? If indeed all approached patients participated, it would reduce non-response bias, but the initial selection method is still a factor.

Authors’ Response: Many thanks for your valued feedback. We believe this unusually high response rate was due to complete anonymity and the design of questionnaire which was simple, quick and easy. Although these characteristics caused the main limitations in the study as we have highlighted in the relevant section.

No exclusion criteria was applied as we approached everyone attending the clinic during that period of time. We have previously reported characteristics from this cohort in detail in another paper (DOI https://doi.org/10.1007/s00520-025-09447-0). But due to the ethics and data protection regulations, we are not able to report it in this paper.

The text states, "We designed the questionnaire..." but makes no mention of whether this questionnaire was validated (e.g., through pilot testing, cognitive interviewing, or comparison with established instruments). An unvalidated questionnaire may have issues with clarity, interpretability, reliability, or validity of the questions, potentially leading to inaccurate or biased responses.

Authors’ Response: We agree with the reviewers’ feedback. We have included the details and limitations of the questionnaire in the revised manuscript.

Your study doesn't explore relationships between variables (e.g., does the impact on sexual health correlate with the effects on relationships? Do those with children pre-cancer report different socio-economic impacts?)

Authors’ Response: We would like to thank the reviewer for their valued feedback. The aim of our study was to highlight the impact of testicular cancer and its treatment on survivors. We are hoping this study will encourage the service provision and awareness among the survivors and healthcare professionals.

You should also provide the number of patients because only the percentages are provided; it is not clear if the number is from the total or from a subgroup

Authors’ Response: We agree with reviewers valued feedback. Amendments have been made as advised.